# Disclosing Potential Key Genes, Therapeutic Targets and Agents for Non-Small Cell Lung Cancer: Evidence from Integrative Bioinformatics Analysis

**DOI:** 10.3390/vaccines10050771

**Published:** 2022-05-12

**Authors:** Md. Parvez Mosharaf, Md. Selim Reza, Esra Gov, Rashidul Alam Mahumud, Md. Nurul Haque Mollah

**Affiliations:** 1Bioinformatics Lab, Department of Statistics, University of Rajshahi, Rajshahi 6205, Bangladesh; parvezstatru@gmail.com (M.P.M.); selim.ru4778@gmail.com (M.S.R.); 2School of Commerce, Faculty of Business, Education, Law and Arts, University of Southern Queensland, Toowoomba, QLD 4350, Australia; 3Centre for High Performance Computing, Joint Engineering Research Centre for Health Big Data Intelligent Analysis Technology, Shenzhen Institutes of Advanced Technology, Chinese Academy of Sciences, Shenzhen 518055, China; 4Department of Bioengineering, Faculty of Engineering, Adana AlparslanTurkes Science and Technology University, Adana 01250, Turkey; egov@atu.edu.tr; 5NHMRC Clinical Trials Centre, Faculty of Medicine and Health, The University of Sydney, Camperdown, NSW 2006, Australia; rashed.mahumud@usq.edu.au

**Keywords:** non-small cell lung cancer, gene expression profiles, molecular signatures, therapeutic targets and agents, integrated bioinformatics approaches

## Abstract

Non-small-cell lung cancer (NSCLC) is considered as one of the malignant cancers that causes premature death. The present study aimed to identify a few potential novel genes highlighting their functions, pathways, and regulators for diagnosis, prognosis, and therapies of NSCLC by using the integrated bioinformatics approaches. At first, we picked out 1943 DEGs between NSCLC and control samples by using the statistical LIMMA approach. Then we selected 11 DEGs (*CDK1*, *EGFR*, *FYN*, *UBC*, *MYC*, *CCNB1*, *FOS*, *RHOB*, *CDC6*, *CDC20*, and *CHEK1*) as the hub-DEGs (potential key genes) by the protein–protein interaction network analysis of DEGs. The DEGs and hub-DEGs regulatory network analysis commonly revealed four transcription factors (*FOXC1*, *GATA2*, *YY1*, and *NFIC)* and five miRNAs (miR-335-5p, miR-26b-5p, miR-92a-3p, miR-155-5p, and miR-16-5p) as the key transcriptional and post-transcriptional regulators of DEGs as well as hub-DEGs. We also disclosed the pathogenetic processes of NSCLC by investigating the biological processes, molecular function, cellular components, and KEGG pathways of DEGs. The multivariate survival probability curves based on the expression of hub-DEGs in the SurvExpress web-tool and database showed the significant differences between the low- and high-risk groups, which indicates strong prognostic power of hub-DEGs. Then, we explored top-ranked 5-hub-DEGs-guided repurposable drugs based on the Connectivity Map (CMap) database. Out of the selected drugs, we validated six FDA-approved launched drugs (Dinaciclib, Afatinib, Icotinib, Bosutinib, Dasatinib, and TWS-119) by molecular docking interaction analysis with the respective target proteins for the treatment against NSCLC. The detected therapeutic targets and repurposable drugs require further attention by experimental studies to establish them as potential biomarkers for precision medicine in NSCLC treatment.

## 1. Introduction

Lung cancer is treated as the leading cause of cancer-related death worldwide among human cancer, which causes the dynamic degradation of the lung [1]. The most common type of bronchial tumor is non-small-cell lung cancer (NSCLC), which accounts for approximately 75% of all lung cancers [2]. The NSCLC is more deadly than the small-cell lung cancer (SCLC), though it grows and spreads slowly compared with the SCLC since it progresses to the advanced stage with few or without any symptoms. Although the targeted therapy has achieved substantial development, the increasing mortality rate associated with lung cancer lays emphasis on both prevention and early detection of lung cancer. Traditional cancer diagnosis methods including histopathology and cytopathologyare practiced in the case of adenocarcinoma, squamous cell carcinoma, and large-cell carcinoma of NSCLC [3,4,5]. The morphological judgment for the tumors has some limitations, including the lack of significant morphological features, which leads to the identification ambiguities [6,7,8,9,10]. Several non-causal risk factors (e.g., smoking, alcohol consumption, and high air pollution) of lung cancer have been detected by several independent studies [11,12,13,14,15]. However, so far, there are no in-depth studies that explore the causal risk factors of NSCLC highlighting their pathogenetic processes and associated candidate drugs for the treatment against NSCLC. The causal risk factors are known as the mutated genes that drive the cancer progression. Usually, non-causal risk factors are assumed to be responsible for genetic mutation and some of them stimulate cancer progression. Cancer-causing mutated genes are utilized for diagnosis, prognosis, and therapies of cancer [16,17]. Moreover, the DNA vaccine is part of a new era of modern therapeutics where the gene-based prophylactic vaccines are being developed [18,19,20]. The plasmid DNA vaccines and viral-vectored vaccines are two types of gene-based vaccines on which many animal trials are being practiced all over the world [21,22]. Therefore, the cancer-causing genes also might be a great therapeutics target for the gene-based DNA vaccine development.

Gene expression profile analysis is now considered as one of the most promising approaches for exploring cancer-causing mutated genes, which yields relevant information for diagnosis, prognosis, and therapies of cancers. [23,24,25,26]. Computationally, mutated genes (potential key genes) are predicted by the analysis of differential gene expression patterns [16,17,23,24,25,26,27,28,29]. Therefore, in this study, an attempt was made to explore NSCLC-causing key genes from the publicly available gene expression profiles, highlighting their functions, pathways, and regulators, which yield relevant information for diagnosis, prognosis, and therapies of NSCLC, by using the integrated bioinformatics approaches.

## 2. Materials and Methods

To reach the goal of this study, we analyzed a publicly available gene expression dataset by using integrated bioinformatics approaches [16,28,29]. The global working flowchart of this study is displayed in Figure 1.

### 2.1. Collection of Gene Expression Profiles for NSCLC

To explore NSCLC-causing key genes, the Affymetrix Human Genome U133 plus 2.0 microarray gene expression dataset was retrieved from the NCBI Gene Expression Omnibus (GEO) database [30] with accession number GSE19804, which contained 60 tumor samples and 60 control samples on 54,675 genes. The dataset was generated by a previous study [31]. The sample unit was aged from 37 to 80 years, with nine different tumor stages (i.e., 1, 1A, 1B, 2, 2A, 2B, 3A, 3B, 4).

### 2.2. Differentially Expressed Genes (DEGs) Identification

At first, the gene expression dataset was normalized for identifying DEGs through the Robust Multi-Array Average (RMA) expression measure and it was implemented by the NCBI-GEO2R (https://www.ncbi.nlm.nih.gov/geo/geo2r/, accessed on 5 October 2021) web-tool. Then, the LIMMA [32] statistical test was utilized to identify the DEGs between NSCLC and control samples. To control the false discovery rate in multiple-testing, the *p*-values were adjusted by Benjamini Hochberg’s [33] method. Both the adjusted *p*-value and log_2_FC values were considered for identifying the upregulated and downregulated DEGs as follows:(1)$$\begin{array}{cc} \mathrm{DEGs}= & \{\begin{array}{c} \mathrm{Upregulated}\;\mathrm{DEGs},\;\mathrm{if}\;\mathrm{adjusted}\;p\;\mathrm{value}<0.001\;\&\;\log_{2}\mathrm{FC}>1\\ \;\mathrm{Downregulated}\;\mathrm{DEGs},\;\mathrm{if}\;\mathrm{adjusted}\;p\;\mathrm{value}<0.001\;\&\;\log_{2}\mathrm{FC}<-1 \end{array} \end{array}$$

### 2.3. DEGs-Set Enrichment Analysis

The bioinformatics resources, Database for Annotation, Visualization and Integrated Discovery (DAVID) (version v6.8) [34,35] was utilized to discern molecular function, biological process, and molecular pathway annotations related to the identified DEGs. Besides, the KEGG pathways identification was conducted through the Kyoto Encyclopedia of Genes and Genomes (KEGG) pathway database [36,37,38]. For statistical significance, the adjusted *p*-value < 0.05 was considered, determined from Fisher Exact test and Benjamini–Hochberg’s correction was used for the multiple testing correction techniques.

### 2.4. Protein-Protein Interaction Network Analysis of DEGs

The STRING database [39] was used to construct the protein–protein interaction (PPI) network of the proteins encoded by DEGs. The STRING database uses a score combiner depending on the product of probabilities [40]. To visualize and perform topological analyses of the PPI network, the NetworkAnalyst [41] was utilized. The topological analysis was applied to determine hub-DEGs/proteins through the CytoHubba plugin [42] in Cytoscape 3.8.2 using degree (connectivity) and betweenness metrics simultaneously [43]. The minimum degree of 10 was considered as the cut off criterion in CytoHubba. Furthermore, the Molecular Complex Detection (MCODE), a novel clustering algorithm [44] along with the CytoHubba was used to identify the sub-modules from the PPI network. The top-scored modules are presented in this analysis.

### 2.5. Mutation Analysis of Hub-DEGs

To investigate the genomic alterations/mutations of the hub-genes, the online cBioPortal (https://www.cbioportal.org, accessed on 28 March 2022) was used over the NSCLC datasets of the server [45,46]. The OncoPrint output was used to represent the most important alteration frequency of genes.

### 2.6. Physicochemical Properties of Hub Proteins

The physicochemical properties of the detected hub proteins were reported from the online tool ProtParam (https://web.expasy.org/protparam/, accessed on 10 November 2021), which allows the computation of various physical and chemical parameters for a given protein. The physiochemical properties of molecular weight, theoretical pI, extinction coefficient, instability index, aliphatic index, and grand average of hydropathicity (GRAVY) were checked for the reported hub protein in this study.

### 2.7. Regulatory Biomolecules Selection

To explore transcriptional and post-transcriptional regulators of DEGs, we performed TFs-DEGs and miRNA-DEGs interaction network analysis, respectively. The TarBase and miRTarBase [47,48] databases were used to identify the significant miRNAs. The JASPAR database [49] retrieved the key regulatory transcription factors (TFs). The entire analysis was conducted through the NetworkAnalyst [41].

### 2.8. Cross-Validation and Evaluation of the Performance of Reported Biomolecules

At first, patients were divided into a low-risk group (control group) and high-risk group (SCLC group) in the SurvExpress online server [50]. Then, the differences between the risk groups from the expression levels of hub-DEGs were investigated by using box plots and survival probability curves.The statistical significance of the differences in the box plots were evaluated through the *t*-test. Survival signatures of the reported biomolecules were evaluated through Kaplan–Meier plots, and a log-rank *p*-value < 0.01 for the statistical significance in all survival analyses.

### 2.9. Drug Repositioning

The hub-DEGs-guided probable drugs or drug candidate molecules were retrieved through the online drug-repositioning tool and database Connectivity Map (CMap) [51]. This is an integrative platform that accumulates the information of the drug or drug candidate molecules from published data sources in clinical experimental stages, investigational stages, and approved for treatment stages. Furthermore, the molecular docking simulation study [52] was conducted for the target biomolecules with the repositioned drug to identify the best-fitted position with binding affinity. The highest docking score with the best-fit pose was considered for the drug–protein interaction affinity. An important type of molecular docking is protein–ligand docking because of its therapeutic applications in modern structure-based drug design [52]. Here, have performed some vital protein ligand docking and studied the interacting amino acids of the same complex. The 3D structure of the target proteins was obtained from Protein Data bank (PDB). The chemical structure of drugs was retrieved from PubChem database (https://pubchem.ncbi.nlm.nih.gov/, accessed on 5 December 2021). All generated chemical compound structures were energy minimized by the MMFF94 force field [53]. For the binding sites, predictions of target proteins were analyzed through 3DLigandSite—Ligand-binding site prediction Server [54]. Docking analysis was carried out using Autodock 4.2 [55] and AutoDock Vina [56]. The interactions like Hydrogen Bonding and other non-bonded terms between all drug and target proteins were carried out using the Accelrys Discovery Studio Visualizer software [57].

## 3. Results

### 3.1. Differentially Expressed Genes (DEGs) Identification

At first, we normalized the genes expression profiles by using RMA. Then, we analyzed the normalized dataset by the statistical LIMMA approach and isolated 1943 DEGs between NSCLC and control samples with the cutoff at adjusted *p*-value < 0.001 and |log_2_FC| < 1 (Figure 2A). Among those, 1367 DEGs were upregulated, and the remaining 576 DEGs were downregulated (Figure 2B). Further analysis was conducted based on these DEGs.

### 3.2. Protein-Protein Interaction Analysis

The PPI network analysis was conducted to reveal the central highly connected proteins which are called hub-DEGs, or proteins or key genes/proteins based on the degree measures (Figure 3) through Cytoscope 3.7.2 with CytoHubba. The degree was considered as ≥10 along with the other default parameters. The proposed top hub proteins are CDK1, EGFR, FYN, UBC, MYC, CCNB1, FOS, RHOB, CDC6, CDC20, and CHEK1, which could be the main proteins in the NSCLC pathogenesis mechanism. By using the MCODE algorithm, 19 sub-network modules were selected considering the default parameters such as node score cutoff of 0.2, K-Core value of 2, and maximum depth from the seed node of 100 along with the other default parameters. Based on the score, the top four modules are represented in Figure 4 and details of analysis results are provided in Appendix A. The sub-modules were checked and the presence of the proposed hub proteins was found. The presence of the hub proteins indicates that these are more reliable to treat as potential therapeutic targets.

### 3.3. Mutation Analysis of Hub-DEGs

The genomic alteration/mutation analysis of 11 hub-DEGs revealed that the *EGFR*, *MYC*, and *CHEK1* genes had 12%, 8%, and 1.3% genomic alteration/mutation over the four lung cancer studies. Other genes were consistent among the studies. For details of the genomic alteration/mutation summary, see Appendix A.

The physicochemical properties of the identified hub proteins are reported in this study. These properties are essential for deeper investigation of the significant biomolecules. The EGFR protein had the highest molecular weight (MW) of 134,277.4 kda, where the UBC reflected the lowest 18,006.82 kda MW. The isoelectric point ranged from 4.77 (FOS) to 9.64 (CDC6) among the reported hub proteins. The detailed information is summarized in Table 1.

### 3.4. Biological Importance of DEGs

DAVID (version v6.8) revealed the molecular function, biological process, and molecular pathway annotations of the identified DEGs through the gene over-representation analysis. The significant GO terms were retrieved, which included the biological processes, molecular function, and cellular components (Table 2). The significant GO terms are summarized and presented in Table 2 for upregulated and downregulated genes separately. The significant functional pathways obtained from the KEGG Pathway analysis are also shown in Figure 5 for the hub-DEGs. The pathways in cancer, cytokine–cytokine receptor interaction, chemokine signaling pathway, cell-adhesion molecules (CAMs), cAMP signaling pathway, MAPK signaling pathway, and TNF signaling pathway are the significant pathways shared by the upregulated genes (Figure 5A). The metabolic pathways, cell cycle, PI3K-Akt signaling pathway, focal adhesion, and ECM-receptor interaction key pathways are exhibited by the downregulated genes (Figure 5B).

### 3.5. Regulatory Transcriptional/Post Transcriptional Candidates in in NSCLC

The TFs-DEGs interaction network and the miRNA-DEGs interaction network revealed the substantial TFs and the miRNAs (Figure 6) that may significantly regulate the DEGs. The transcription factors (*FOXC1*, *GATA2*, *YY1*, *E2F1*, *FOXL1*, *NFIC*, *NFKB1*, *PPARG*, *TFAP2A*, *USF2*) and miRNA (miR-335-5p, miR-26b-5p, miR-16-5p, miR-124-3p, miR-92a-3p, miR-7b-5p, miR-93-5p, miR-17-5p, miR-155-5p) were selected as the key transcriptional and post-transcriptional regulatory biomolecules of DEGs. Furthermore, the interaction network of hub proteins with TFs and miRNA were constructed (Figure 7). The hub-proteins versus TFs interaction network reflected four TFs (*FOXC1*, *GATA2*, *YY1*, and *NFIC)* as the key regulatory TFs of the drug target hub-DEGs/proteins (Figure 7A). On the other hand, five miRNAs (miR-335-5p, miR-26b-5p, miR-92a-3p, miR-155-5p, and miR-16-5p) were found as the key regulatory miRNAs of hub-DEGs/proteins (Figure 7B). These regulatory biomolecules were also found from the interaction network analysis of DEGs-TF and all DEGs-miRNA, respectively (Figure 6).

### 3.6. Risk Discrimination Performance of Reporter Biomolecules

The risk discrimination performance and the differential expression pattern were observed by the online gene validation website SurvExpress. The analysis was conducted through the TCGA Lung squamous cell carcinoma survival information for the hub genes and the key transcription factors. The survival curve for the high- and low-risk group and the box plot of their gene expressions are shown in (Figure 8). For both analyses, the prognostic index, log-rank test, and hazard ratio are shown (Figure 8). All hub proteins and reported TFs showed statistically significant performances in terms of survival probabilities in all datasets, in both the high- and low-risk groups.

### 3.7. Drug Repositioning

The identification of the drug candidate molecules through CMap database revealed the repurposed drugs for the top hub drug-target proteins. The CMap database reflected the drug candidate molecules for the submitted hub proteins. Among the top hub proteins, for the *CDK1*, *EGFR*, *FYN*, and *MYC*, we found repurposable drugs in pre-clinical trials, FDA-approved drugs, and those in other experimental stages (Table 3).

The molecular docking analysis for the FDA-approved, launched drugs with the hub proteins was conducted. The best pose with the highest docking score was considered to select the drug–protein interaction. The potential repositioned drug candidates need deeper attention for further experimental validation, which leads to the development of more efficient therapy for NSCLC treatment. The molecular docking analysis results are summarized in Figure 9, where (i) indicates the protein–drug complex and (ii) indicates the 2D diagram with interacting amino acid. For the Dinaciclib–*CDK1* complex, interaction in the substrate-binding site (SBS-1) of *CDK1* generated a binding-free energy of −9.3 Kcal/mol. Residues such as THR14, TYR15, VAL18, LYS33, GLN132, ASN133, ALA145, ASP146, and VAL165 surround the amino acid and THR14, GLN132, GLN132, ASN133, and ASP146 are involved in the hydrogen-bond interaction while the other surrounding amino acid residues are involved in hydrophobic interactions (Figure 9A). The docking simulation of EGFR inhibitor was performed with three compounds, including Afatinib, Erlotinib, and Gefitinib (Figure 9B–D). The highest affinity for substrate binding sites (SBS-2), with a binding free energy of −9.0 Kcal/mol, was found for Afatinib in the EGFR open conformation model, and binding-free energies of −8.5 Kcal/mol and −8.2 Kcal/mol were found for for Erlotinib and Gefitinib compounds in EGFR conformations respectively. Therefore, the chemical compound of Afatinib was strongly bound with EGFR conformation. LEU718, LYS745, MET793, CYS797, ARG841, ASN842, ASP855, and LEU858 are the surrounding residues for the Afatinib–*EGFR* complex. MET793, ASN842, ASP855, and LEU718 are involved in the hydrogen-bond interaction, while the other surrounding amino acids such as LYS745, LEU718, LEU858, and ARG841, CYS797, and ARG841 are involved in Pi–Cation, Alkyl, and Pi–Alkyl interactions respectively. The docking simulation of *FYN* inhibitor was performed with two compounds including Bosutinib and Dasatinib (Figure 9E,F). The highest affinity for SBS-3, with a binding-free energy of −7.1 Kcal/mol, was found for Bosutinib in FYN conformation, and a binding-free energy of −6.9 Kcal/mol was found for the Dasatinib—EGFR complex. Therefore, the compound of Bosutinib was strongly bound with the FYN conformation. Trp149, Tyr150, Arg176, Leu224, and Gln225 are surrounding residues for the Bosutinib–FYN complex. TRP149, GLN225, and ARG176 are involved in the hydrogen-bond interaction, while the other surrounding amino acids such as TRP149 and TRP149, and TRP149, TYR150, and LEU224 are involved in C-H and Pi-Orbital’s interactions respectively.

For the TWS-119–MYC complex, the interaction in SBS-4 of MYC generated a binding-free energy of −7.9 Kcal/mol. Residues such as Ser952, Val953, Glu956, Arg254, His258, and Gln261 are surrounding amino acids and GLN261, GLU956, and HIS258 are involved in the hydrogen-bond interaction while the other surrounding amino acid residues are involved in hydrophobic interactions (Figure 9G). The ultimate potential of the drugs with the molecular signatures of the NSCLC demanded close attention for experimental validation for developing effective and safe medications.

## 4. Discussion

Identification of disease-causing crucial biomarkers may shed light on a deeper understanding of the molecular mechanism of disease [58,59,60,61,62,63]. The present study was conducted to analyze the NSCLC gene expression data to determine the DEGs, extensive molecular pathways, significant hub proteins, and associated regulatory biomolecules in order to pick up the potential therapeutic targets for NSCLC through a multi-omics data integration framework. Through the gene expression patterns analysis, we identified 1943 DEGs, including 1367 upregulated and 576 downregulated genes. The functional enrichment analysis revealed that the proposed upregulated DEGs are significantly involved with some cancer-causing molecular functions and pathways, including cytokine–cytokine receptor interaction, chemokine signaling pathway, cell-adhesion molecules (CAMs), cAMP signaling pathway, MAPK signaling pathway, TNF signaling pathway, cGMP-PKG signaling pathway, Proteoglycans in cancer, and Rap1 signaling pathway (Figure 5). The downregulated genes are shared metabolic pathways, cell cycle, PI3K-Akt signaling pathway, focal adhesion, ECM-receptor interaction, p53 signaling pathway, and protein digestion and absorption pathways. All of these functions and pathways are significantly related to cancer development and play crucial roles in the NSCLC microenvironment. Recent studies indicated the importance of the tumor microenvironment as a decisive factor in tumorigenesis in various cancers [64,65,66,67,68]. Therefore, the physicochemical properties will be helpful to explore the further analysis of the reported proteins as a therapeutic target for NSCLC.

To detect the basic mechanism of disease, the protein–protein interaction network analysis is becoming a promising approach [69]. The PPI network analysis in this study revealed the hub-DEGs’ encoded hub-proteins. The CDK1 is related to the cell cycle activities. Up-regulation of *CDK1* genes may be indicative of poor survival rates and a higher risk for cancer recurrence. The *CDK1* gene is also related to several other cancer diseases [70,71]. The *EGFR* gene is associated with cell growth and had a contribution in lung cancer studied before [72,73]. The study revealed that the growth is suppressed and the radiosensitivity is amplified by the activities of ubiquitin C (*UBC*) in NSCLC cells [74,75,76,77]. The *CDC6*, *CDC20*, and *CHEK1* genes are closely related to the occurrence and development of small-cell lung cancer, and *CHEK1* is treated as a therapeutic target for lung cancer [78]. Eight hub genes (*CDK1*, *EGFR*, *UBC*, *MYC*, *CCNB1*, *RHOB*, *CDC6*, and *CDC20*) have tumor suppressor functions, while five hub genes (*CDK1*, *EGFR*, *FYN*, *UBC*, and *CCNB1*) are protein kinases as well. The MCODE cluster analysis clearly showed that the hub genes were distributed among the distinguished sub network (Figure 4) modules, which provided the strong evidence about the proposed signature biomolecules that these are reliable as therapeutic targets. Thus, the predicted hub-DEGs and relevant information might be useful in early detection of NSCLC. On the other hand, the genomic alteration/mutation analysis of the hub-DEGs reflected that most mutation for the *EGFR* occurred across the four lung cancer studies and was followed by the *MYC* and *CHEK1* genes, since *EGFR* is a highly mutant/altered gene for lung cancer and NSCLC as well [79]. The alteration/mutation frequency revealed that the *EGFR* showed the highest alteration frequency relative to others, including the mutation, where *CHEK1* represented mutation and deletion across the studies (Appendix A), which may be a concern of investigation in future research.

The DEGs and hub-DEGs regulatory network analysis commonly revealed four transcription factors (*FOXC1*, *GATA2*, *YY1*, and *NFIC*) and five miRNAs (miR-335-5p, miR-26b-5p, miR-92a-3p, miR-155-5p, and miR-16-5p) as the key transcriptional and post-transcriptional regulators of DEGs as well as hub-DEGs. A study reported that various tumor-associated genes are regulated by *FOXC1* and maintain several cancer-related pathways [80]. The *GATA2* is treated as a therapeutic target in NSCLC treatment development and it also related to breast and kidney cancer [81,82]. The higher expression pattern of *YY1* transcription factor triggered the patients having larger tumor size, differentiation, higher TNM stage, and lymph node metastasis [83]. The reported TFs are also involved in other cancer diseases [58,59,60,61,62,63]. In various types of cancer tissues, the miR-26b-5p acts as a tumor suppressor [84]. Currently, as one of the diagnostic tools for lung cancer identification, the miR-92a-3p expression measurement is being used [85,86]. The miR-155-5p is significantly associated with a higher risk for progression in adenocarcinoma patients [87,88] and miR-16-5p showed higher expression pattern in NSCLC cells [86].

The prognostic power of the reported biomolecules in discriminating the high- and low-risk conditions were exhibited by using the multivariate survival probability curves and box plots (Figure 8). The survival curves clearly demonstrated that the reported biomolecules played a significant role in patient survival. The box plot of the gene expression data of the molecular candidate also showed clear differences between the high- and low-risk groups (Figure 8B).

Finally, we selected the top-ranked five hub-DEGs-guided candidate drugs from the Connectivity Map (CMap) database (Table 3). Out of the selected drugs, we validated FDA approved six launched drugs (Dinaciclib, Afatinib, Icotinib, Bosutinib, Dasatinib, and TWS-119) by molecular docking simulation with the top-ranked five hub-DEGs-mediated target proteins for the treatment against NSCLC. The drug-target binding affinity scores (less than −7.0 Kcal/mol) suggested that the aforementioned six FDA approved launched drugs might be effective for the treatment against NSCLC. Thus, the findings of this study might be useful resources in prevention and early detection of NSCLC.

## 5. Conclusions

The current study focused on identifying the significant biomolecules along with their molecular mechanisms through integrative bioinformatics analysis. Among 1943 DEGs, 11 DEGs (*CDK1*, *EGFR*, *FYN*, *UBC*, *MYC*, *CCNB1*, *FOS*, *RHOB*, *CDC6*, *CDC20*, and *CHEK1*) were reported as the hub-DEGs/proteins that may play the key roles in NSCLC progression. The DEGs set enrichment analysis with the gene ontology (GO) database showed that DEGs are significantly involved with the cell adhesion, cell division, inflammatory response, signal transduction, protein binding, and plasma membrane extracellular region. The enrichment analysis with the KEGG pathway database showed that DEGs are significantly associated with the metabolic pathways, cell cycle, ECM-receptor interaction, and pathways in cancer. The inevitable regulatory TFs (*FOXC1*, *GATA2*, *YY1*, and *FOXL1*) and miRNA (miR-335-5p, miR-26b-5p, miR-92a-3p, miR-155-5pm and miR-16-5p) were identified as potential regulatory biomarkers for both DEGs and hub-DEGs. The strong prognostic performance of the reported biomolecules was observed between the high- and low-risk groups through the survival curves and box plots. The top-ranked hub-DEG-guided repurposable drug analysis revealed that the Dinaciclib, Afatinib, Icotinib, Bosutinib, Dasatiniband, and TWS-119 might be suggested as novel putative drugs for NSCLC treatment. The molecular docking analysis between the drug-target hub proteins and the repurposed drugs were conducted to investigate their molecular interaction mechanism. Thus, the findings of this study might be useful resources for NSCLC diagnosis, prognosis, and therapies, including gene-based DNA-vaccine development.

## Figures and Tables

**Figure 1 vaccines-10-00771-f001:**
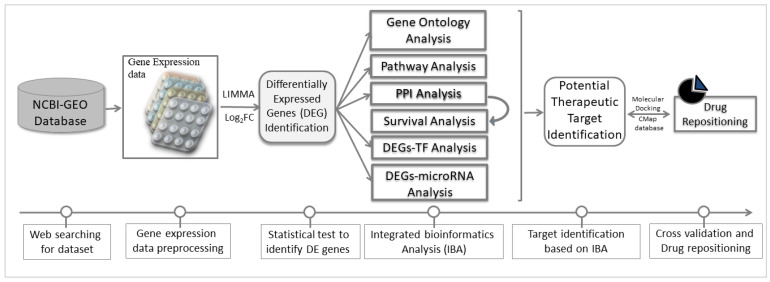
The schematic diagram of the integrative bioinformatics analysis of this study.

**Figure 2 vaccines-10-00771-f002:**
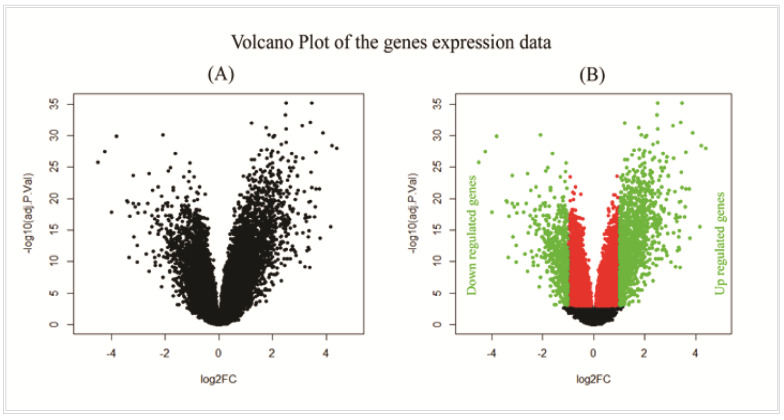
Gene expression profile of microarray data. (**A**) The volcano plot which represents the scatter plot of log_2_FC values versus −log_10_(adjusted *p*-values). (**B**) The volcano plot highlighting DEGs, where green bullets represent the upregulated (adjusted *p*-value < 0.001 and log_2_FC > 1) and downregulated (adjusted *p*-value < 0.001 and log_2_FC < −1) DEGs selected based on the described criteria.

**Figure 3 vaccines-10-00771-f003:**
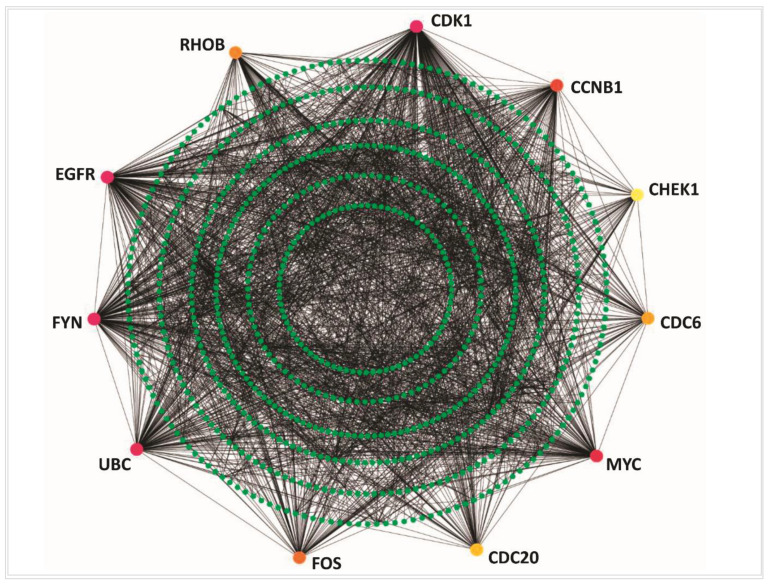
NSCLC-specific protein–protein interaction network. The redder color represents the higher degree measured by CytoHubba. The hub-DEGs are represented only with the different colors in the PPI. Green nodes represent the associated proteins.

**Figure 4 vaccines-10-00771-f004:**
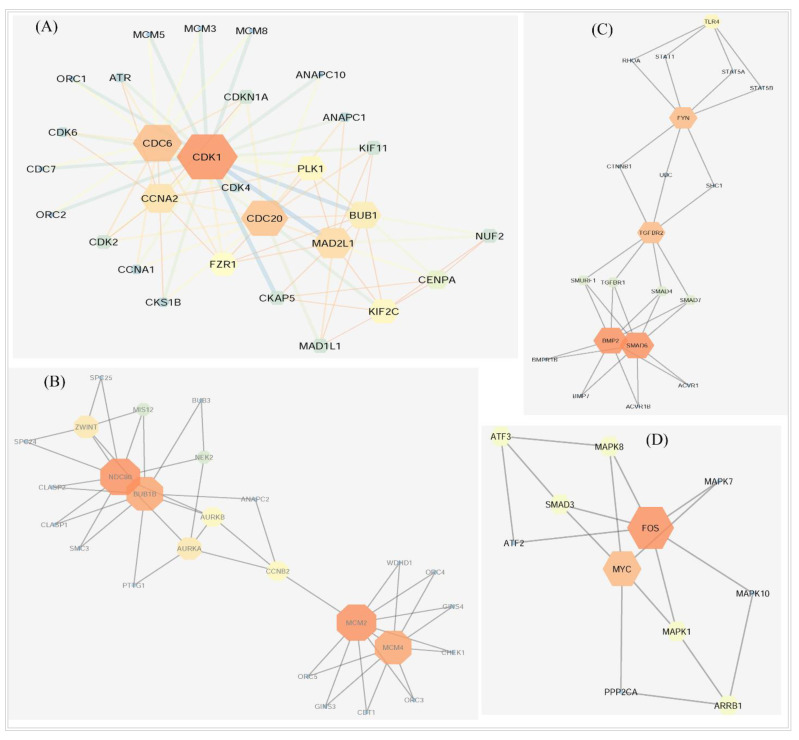
The first four sub networks based on score, identified by the MCODE algorithm. The scores of 6.071, 3.76, 3.684, and 3.4 were exhibited by the (**A**) first, (**B**) second, (**C**) third, and (**D**) forth sub modules, respectively.

**Figure 5 vaccines-10-00771-f005:**
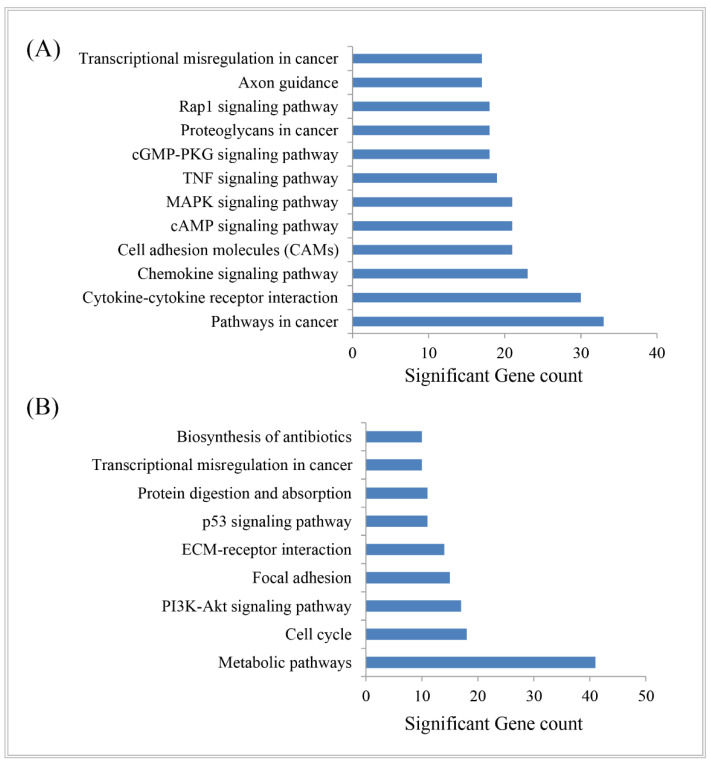
The KEGG pathways (**A**) for upregulated DEGs and (**B**) downregulated DEGs.

**Figure 6 vaccines-10-00771-f006:**
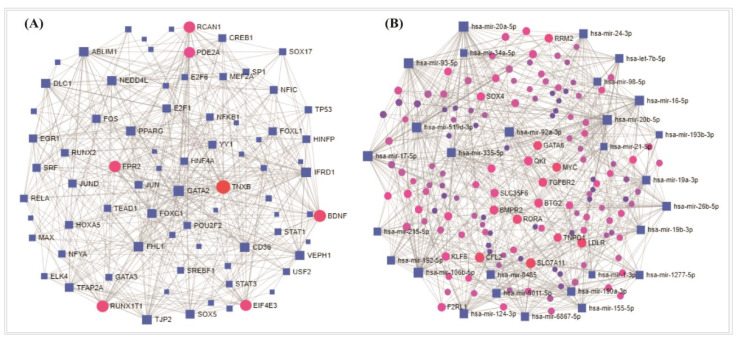
(**A**) The TFs-DEGs interaction network and (**B**) the miRNA-DEGs interaction network. The TFs and miRNAs are marked as blue-shape square in the interactions. The larger square means a higher degree of connectivity among the nodes. The circle-shaped nodes represent the DE genes.

**Figure 7 vaccines-10-00771-f007:**
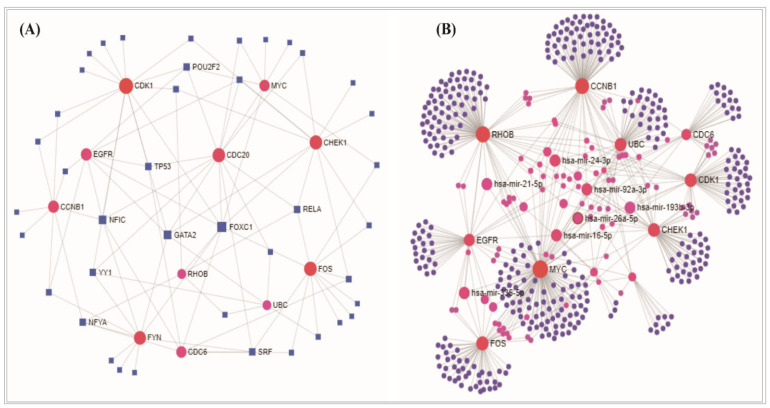
(**A**) The hub proteins–TFs interaction network, and the TFs are marked as blue-shaped square in the interactions. (**B**) The hub proteins–miRNA interaction network, and the hub proteins are marked as red circles in interaction network. The larger significant miRNAs are labeled and marked as pink-colored circles.

**Figure 8 vaccines-10-00771-f008:**
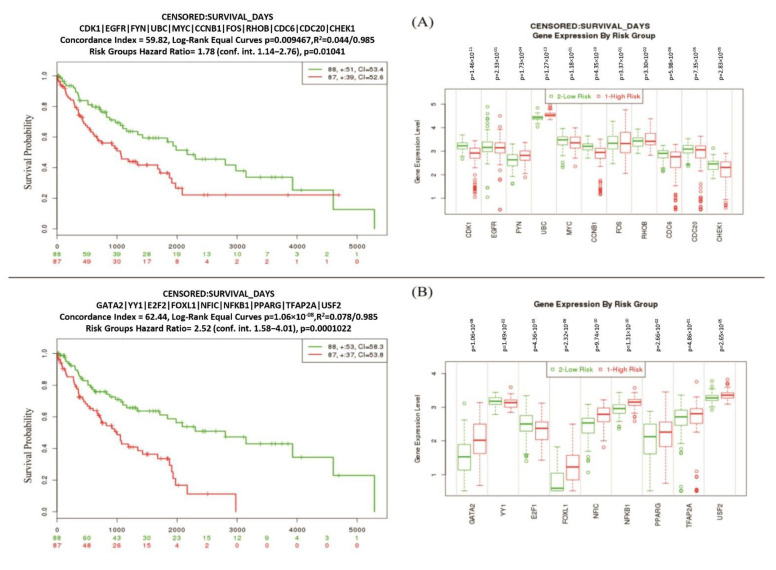
The risk group discrimination performance by the multivariate survival probability curves (left) and box plots (right) based on (**A**) hub-DEGs/proteins and (**B**) key TFs (transcription factors) proteins.

**Figure 9 vaccines-10-00771-f009:**
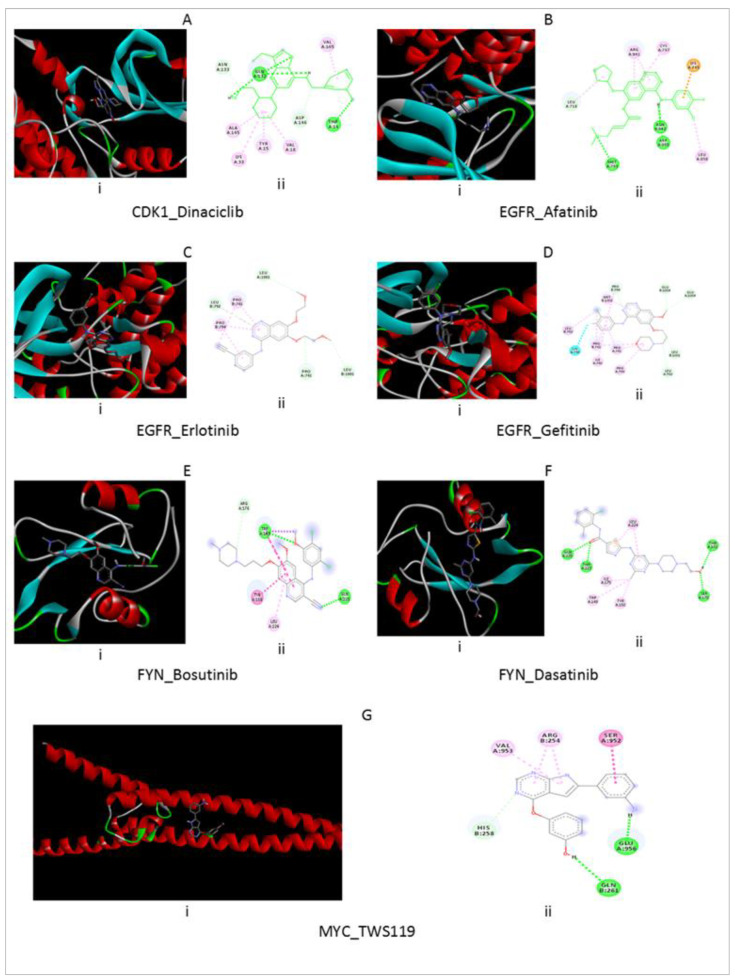
The molecular docking poses for the selected repurposed drugs and potential target proteins. The figure showed the best docking pose between protein and drug, like in (**A**) between CDK1-Dinaciclib; in (**B**) between EGFR-Afatinib; in (**C**) between EGFR-Erlotinib; in (**D**) between EGFR-Gefitinib; in (**E**) between FYN-Bosutinib; in (**F**) between FYN-Dasatinib and in (**G**) between MYC-TWS119 respectively.

**Table 1 vaccines-10-00771-t001:** The physicochemical properties of the reported hub proteins.

Hub Protein’s Name	Number of Amino Acids	Molecular Weight (kda)	Theoretical pI	Number of Negatively Charged Residues (Asp + Glu)	Number of Positively Charged Residues (Arg + Lys)	* Extinction Coefficient	Instability Index	Aliphatic Index	Grand Average of Hydropathicity (GRAVY)
**CDK1**	297	34,095.45	8.38	37	39	42,860	39.26	97.78	−0.281
**EGFR**	1210	134,277.4	6.26	138	126	128,890	44.59	80.74	−0.316
**FYN**	537	60,761.9	6.23	68	63	94,240	36.41	75.36	−0.489
**UBC**	158	18,006.82	8.87	18	22	29,700	45.78	72.91	−0.533
**MYC**	439	48,804.08	5.33	64	51	29,505	92.23	66.42	−0.772
**CCNB1**	433	48,337.43	7.09	52	52	30,620	50.59	90.09	−0.239
**FOS**	380	40,695.41	4.77	51	33	21,930	78.82	65.32	−0.369
**RHOB**	196	22,123.39	5.1	32	26	21,930	46.35	87.96	−0.26
**CDC6**	560	62,720.28	9.64	58	91	20,940	48.57	94.89	−0.383
**CDC20**	499	54,722.59	9.33	42	54	106,255	47.72	76.31	−0.483
**CHEK1**	476	54,433.57	8.5	61	66	76,485	42.26	84.75	−0.459

Note: * Extinction coefficients are in units of M^−1^ cm^−1^, at 280 nm measured in water.

**Table 2 vaccines-10-00771-t002:** The functional enrichment analysis of the DEGs to clarify the gene ontology terms in the NSCLC disease. The top GO terms are summarized and presented here.

** *Upregulated Genes* **				
**GO Term**		**Number of Genes**	**Coverage (%)**	***p*-Value**
GOTERM_BP_DIRECT				
GO:0001525 angiogenesis		40	4.27	1.77 × 10^−12^
GO:0007155 cell adhesion		59	6.3	1.28 × 10^−11^
GO:0006954 inflammatory response		50	5.3	2.33 × 10^−10^
GO:0007166 cell-surface receptor signaling pathway		41	4.4	2.97 × 10^−10^
GO:0006955 immune response		49	5.2	2.29 × 10^−8^
GO:0032496 response to lipopolysaccharide		26	2.8	2.80 × 10^−7^
GO:0006935 chemotaxis		22	2.3	3.17 × 10^−7^
GO:0007165 signal transduction		94	10.0	5.91 × 10^−7^
GOTERM_CC_DIRECT				
GO:0005886 plasma membrane		295	31.5	8.30 ×10^−16^
GO:0005576 extracellular region		145	15.5	1.69 ×10^−14^
GO:0005615 extracellular space		127	13.5	3.91× 10^−14^
GO:0045121 membrane raft		34	3.6	6.18 × 10^−10^
GO:0070062 extracellular exosome		185	19.7	9.29 × 10^−7^
GO:0009986 cell surface		52	5.5	2.02 × 10^−6^
GO:0005925 focal adhesion		41	4.4	3.45 × 10^−6^
GO:0016021 integral component of membrane		297	31.7	2.91 × 10^−5^
GOTERM_MF_DIRECT				
GO:0008201 heparin binding		29	3.1	1.15 × 10^−9^
GO:0030246 carbohydrate binding		27	2.9	1.36 × 10^−6^
GO:0005178 integrin binding		19	2.0	1.46 × 10^−6^
GO:0005509 calcium ion binding		59	6.3	2.60 × 10^−5^
GO:0051015 actin filament binding		19	2.0	3.86 × 10^−5^
GO:0004872 receptor activity		25	2.7	7.30 × 10^−5^
GO:0005515 protein binding		460	49.1	8.91 × 10^−5^
GO:0003779 actin binding		28	3.0	2.41 × 10^−4^
** *Down Regulated Genes* **				
**GO Term**		**Number of Genes**	**Coverage (%)**	***p*-Value**
GOTERM_BP_DIRECT				
GO:0030574	collagen catabolic process	15	3.4	1.70 × 10^−10^
GO:0007067	mitotic nuclear division	26	5.9	7.35 × 10^−10^
GO:0051301	cell division	29	6.5	1.30 × 10^−8^
GO:0007062	sister chromatid cohesion	14	3.2	7.36 × 10^−7^
GO:0030198	extracellular matrix organization	19	4.3	7.37 × 10^−7^
GO:0000082	G1/S transition of mitotic cell cycle	13	3.0	4.17 × 10^−6^
GO:0030199	collagen fibril organization	8	1.8	2.75 × 10^−5^
GO:0001649	osteoblast differentiation	12	2.7	2.90 × 10^−5^
GO:0000281	mitotic cytokinesis	7	1.6	4.50 × 10^−5^
GO:0006508	proteolysis	27	6.1	1.12 × 10^−4^
GOTERM_CC_DIRECT				
GO:0005615	extracellular space	63	14.2	5.08 × 10^−8^
GO:0070062	extracellular exosome	101	22.8	1.18 × 10^−6^
GO:0005578	proteinaceous extracellular matrix	21	4.7	3.05 × 10^−6^
GO:0000777	condensed chromosome kinetochore	12	2.7	3.95 × 10^−6^
GO:0005581	collagen trimer	12	2.7	6.85 × 10^−6^
GO:0030496	midbody	14	3.2	6.95 × 10^−6^
GO:0005576	extracellular region	64	14.4	1.01 × 10^−5^
GO:0005819	spindle	12	2.7	9.10 × 10^−5^
GOTERM_MF_DIRECT				
GO:0004222	metalloendopeptidase activity	13	2.9	7.55 × 10^−6^
GO:0004252	serine-type endopeptidase activity	19	4.3	1.56 × 10^−5^
GO:0005201	extracellular matrix structural constituent	10	2.2	1.57 × 10^−5^
GO:0042802	identical protein binding	32	7.2	6.18 × 10^−4^
GO:0019901	protein kinase binding	19	4.3	0.0019
GO:0005524	ATP binding	51	11.5	0.0021

**Table 3 vaccines-10-00771-t003:** The repurposed drugs that were found from the CMap database.

Target Proteins	Name of Drug	Mechanism of Action	Phase
*CDK1*	aminopurvalanol-a	CDK inhibitor, tyrosine kinase inhibitor	Pre-clinical
	BMS-265246	CDK inhibitor	Pre-clinical
	CDK1-5-inhibitor	CDK inhibitor, glycogen synthase kinase inhibitor	Pre-clinical
	CGP-60474	CDK inhibitor	Pre-clinical
	CGP-74514	CDK inhibitor	Pre-clinical
	CHIR-99021	glycogen synthase kinase inhibitor	Pre-clinical
	dinaciclib	CDK inhibitor	Phase 3
	indirubin-3-monoxime	CDK inhibitor, glycogen synthase kinase inhibitor	Pre-clinical
	JNJ-7706621	CDK inhibitor	Pre-clinical
	kenpaullone	CDK inhibitor, glycogen synthase kinase inhibitor	Pre-clinical
	olomoucine	CDK inhibitor	Pre-clinical
	PF-573228	focal adhesion kinase inhibitor	Pre-clinical
	PHA-767491	CDC inhibitor	Pre-clinical
	purvalanol-a	CDK inhibitor	Pre-clinical
	Ro-3306	CDK inhibitor	Pre-clinical
	SU9516	CDK inhibitor	Pre-clinical
	1-azakenpaullone	glycogen synthase kinase inhibitor	Pre-clinical
	8-hydroxy-DPAT	serotonin receptor agonist	Pre-clinical
*EGFR*	afatinib	EGFR inhibitor	Launched
	brigatinib	ALK tyrosine kinase receptor inhibitor, EGFR inhibitor	Launched
	erlotinib	EGFR inhibitor	Launched
	gefitinib	EGFR inhibitor	Launched
	icotinib	EGFR inhibitor	Launched
	lapatinib	EGFR inhibitor	Launched
	lidocaine	histamine receptor agonist	Launched
	olmutinib	EGFR inhibitor, Bruton’s tyrosine kinase (BTK) inhibitor	Launched
	osimertinib	EGFR inhibitor	Launched
	vandetanib	EGFR inhibitor, RET tyrosine kinase inhibitor, VEGFR inhibitor	Launched
*FYN*	bosutinib	Abl kinase inhibitor, Bcr-Abl kinase inhibitor, src inhibitor	Launched
	dasatinib	Bcr-Abl kinase inhibitor, ephrin inhibitor, KIT inhibitor, PDGFR tyrosine kinase receptor inhibitor, src inhibitor, tyrosine kinase inhibitor	Launched
*MYC*	TWS-119	glycogen synthase kinase inhibitor	Pre-clinical

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
