# Peer review of "Disclosing Potential Key Genes, Therapeutic Targets and Agents for Non-Small Cell Lung Cancer: Evidence from Integrative Bioinformatics Analysis"

_vaccines, 2022, doi:10.3390/vaccines10050771_

Round 1

Reviewer 1 Report

The authors demonstrated novel genes for NSCLC target via bioinformatics analysis. Its approach is novel and interesting, however, some point should be revised. The specific comments are as following:
Abstract: Too long! Main issues should be presented only.
Introduction: The goal of this study should be emphasized. And, possible results or hypothesis should be presented. Difference with previous study? Why you want to publish this article in Vaccine journal?

Method: Many analysis (formula, geometric distribution format, and et al) was similar with previous studies, therefore, short description with reference is possible.

Discussion: Discussion part should give more information about authors results. Previous results in other articles were described extremely. Possible mechanism of its effect and clinical value of these data should be presented more.

Reference style should be checked.

Author Response

Reviewer 1

General Comment:

Comments and Suggestions for Authors

The authors demonstrated novel genes for NSCLC target via bioinformatics analysis. Its approach is novel and interesting; however, some point should be revised. The specific comments are as following:

Response: Thank you very much for your interest with some important comments and suggestions on our research. We updated the manuscript according to your specific comments.

Comments:

Abstract: Too long! Main issues should be presented only.

Response: The abstract has been updated accordingly by keeping the main issues in the revised manuscript (MS).

Introduction: The goal of this study should be emphasized. And possible results or hypothesis should be presented. Difference with previous study? Why you want to publish this article in Vaccine journal?

Response: Thank you so much for your important suggestions. We re-wrote the introduction section according to your suggestion. Gene-based vaccine development is a promising research topic. We also mentioned this issue in the introduction section according to your suggestion.  

Method: Many analysis (formula, geometric distribution format, and et al) was similar with previous studies, therefore, short description with reference is possible.

Response: Thanks for this suggestion. We have revised the methodology section according to your suggestion.

Discussion: Discussion part should give more information about authors results. Previous results in other articles were described extremely. Possible mechanism of its effect and clinical value of these data should be presented more.

Response: Thanks for this suggestion. We have updated the discussion section according to your suggestion in the revised manuscript.

Reference style should be checked.

Response: The reference style has been checked and updated accordingly.

Reviewer 2 Report

Journal: Vaccines (ISSN 2076-393X)

Manuscript ID: vaccines-1647540

Title

Discloses potential novel genes, therapeutic targets and agents for non-small cell lung cancer: An evidence from integrative bioinformatics analysis

Reviewer comments

The authors analysed the NSCLC gene expression data to find out the DEGs, molecular pathways, significant hub proteins, associated regulatory biomolecules in order to pick up the potential therapeutic targets for NSCLC through a multi-omics data integration framework. The aim of the study was to individualise the more informative and signature biomolecules hub genes, TFs as well as miRNA along with their biological and cellular mechanism.

The manuscript is organized and reasonably written, but needs to undergo English language proofreading. While the data are of value, the manuscript itself could benefit from some improvement and clarification.

  1. Spell check:
    1. The manuscript title should be rephrased to make it clearer. Starting with Discloses is not fluent…
    2. Line 59: lung cancer lays emphasise
    3. Line 92: can be sutilised to identify the significant therapeutic bio…
    4. Line 111: sutilised in this study.
    5. Line 122: The gene expression dataset was snormalised for identifying DEGs
    6. Line 291: The top GO terms are ssummarised and presented here.
    7. There are others. The manuscript should be edited for the English language…

  1. Can you analyse if the identified key signalling and regulatory molecules are associated with NSCLC, (hub-DEGs, their significant regulatory transcription factors and their important miRNA): Are these key molecules predicted to be associated with small-cell lung cancer or other cancer types? How unique are these molecules to NSCLC?

  1. The sample unit used in the study was aged from 37-to 80 years, with nine different tumour stages, thus could provide an opportunity to predict cancer survivor outcomes. Are there molecules from these data sets of genes, miRNA predictions that could be associated with early or late diagnosis of NSCLC?

  1. Several risk factors were declared the key risk factors of lung cancer in this study. Could you speculate on potential molecules from this data set of genes, miRNA, that could be predicted to be associated with some reported risk factors?

  1. From these data sets, are there molecules that could be predicted to potentially reduce the mortality rate in lung cancer? How would this analysis be expanded to explore the molecular signatures for therapeutic improvement?

  1. Mutation analysis of the novel therapeutic target biomolecules (i.e. proteins, TF, miRNA). Mutation analysis is a common way to detect therapeutic sensitizing and resistant mutations. This will also allow for a precise prognosis and diagnosis, as well as personalized therapies tailored to specifically meet with the nine different tumour stages of each patient.

Are there known mutations or predicted alterations that could affect the prognostic capabilities of the therapeutic target biomolecules?

  1. How relevant and significant are the differences between the low and high-risk groups?

Author Response

Reviewer 2

Comments and Suggestions for Authors

Journal: Vaccines (ISSN 2076-393X)

Manuscript ID: vaccines-1647540

Title: “Discloses potential novel genes, therapeutic targets and agents for non-small cell lung cancer: An evidence from integrative bioinformatics analysis”

Reviewer comments

General comment:

The authors analysed the NSCLC gene expression data to find out the DEGs, molecular pathways, significant hub proteins, associated regulatory biomolecules in order to pick up the potential therapeutic targets for NSCLC through a multi-omics data integration framework. The aim of the study was to individualize the more informative and signature biomolecules hub genes, TFs as well as miRNA along with their biological and cellular mechanism.

The manuscript is organized and reasonably written but needs to undergo English language proofreading. While the data are of value, the manuscript itself could benefit from some improvement and clarification.

Response: Thanks for your appreciation with important comments for the improvement of the manuscript.

Comments:

  1. Spell check:
    1. The manuscript title should be rephrased to make it clearer. Starting with Discloses is not fluent…
    2. Line 59: lung cancer lays emphasise
    3. Line 92: can be sutilised to identify the significant therapeutic bio…
    4. Line 111: sutilised in this study.
    5. Line 122: The gene expression dataset was snormalised for identifying DEGs
    6. Line 291: The top GO terms are ssummarised and presented here.
    7. There are others. The manuscript should be edited for the English language…

Response: Thanks for picking the grammatical errors. We have checked the MS thoroughly and corrected the grammatical errors and spelling mistakes. Please find them in the updated MS.

  1. Can you analyse if the identified key signaling and regulatory molecules are associated with NSCLC, (hub-DEGs, their significant regulatory transcription factors and their important miRNA): Are these key molecules predicted to be associated with small-cell lung cancer or other cancer types? How unique are these molecules to NSCLC?

Response: Our predicted key biomolecules (Hub-protein, TFs, miRNAs) were significantly associated with NSCLC. The literature review showed that they are also associated with some other types of cancers. Please see the discussion section in the revised manuscript.  

  1. The sample unit used in the study was aged from 37-to 80 years, with nine different tumour stages, thus could provide an opportunity to predict cancer survivor outcomes. Are there molecules from these data sets of genes, miRNA predictions that could be associated with early or late diagnosis of NSCLC? 

Response: Yes, we think that the predicted key biomolecules can be utilized for early diagnosis of NSCLC. We mention this issue in the introduction and discussion section.

  1. Several risk factors were declared the key risk factors of lung cancer in this study. Could you speculate on potential molecules from this data set of genes, miRNA, that could be predicted to be associated with some reported risk factors? 

Response: Thanks for your nice question. We introduced both causal and non-causal risk factors of cancers including NSCLC. In our study, we explored both causal (key genes) and non-causal (TFs, miRNAs) molecular risk factors of NSCLC. The literature review supported our predicted risk factors of NSCLC.

  1. From these data sets, are there molecules that could be predicted to potentially reduce the mortality rate in lung cancer? How would this analysis be expanded to explore the molecular signatures for therapeutic improvement? 

Response: Yes, the finding of our study might be useful to reduce the mortality rate from NSCLC. Our predicted key biomolecules can be utilized for early diagnosis of NSCLC and identified repurposable drugs can be effective for the treatment against NSCLC. Thus, mortality rate can be reduced. However, our in-silico findings require further attention by wet-lab experimental studies to established them as the potential biomarkers and effective candidate drugs for the treatment against NSCLC.

  1. Mutation analysis of the novel therapeutic target biomolecules (i.e., proteins, TF, miRNA). Mutation analysis is a common way to detect therapeutic sensitizing and resistant mutations. This will also allow for a precise prognosis and diagnosis, as well as personalized therapies tailored to specifically meet with the nine different tumor stages of each patient. Are there known mutations or predicted alterations that could affect the prognostic capabilities of the therapeutic target biomolecules?

Response: Thank you very much for your important suggestion. We have checked the genomic alteration/mutation of the 11 hub-DEGs across the lung cancers in the cBioPortal database (https://www.cbioportal.org/). The EGFR, MYC and CHEK1 genes showed the genetic alteration/mutation across the cancer studies. We have updated the methodology, results and discussion section according to our findings in the updated MS. Please check in the Materials and methods sub-section 2.4, in result section and discussion section. The details report has been provided in supplementary figure 1.

  1. How relevant and significant are the differences between the low and high-risk groups?

Response: Our key biomolecules showed significant survival difference between the low and high risks group for lung cancer in the SurvExpress database. However, box plot analysis showed that only two key genes EFFR and Fox out of 11 shows insignificant difference between the low- and high- risks group with the independent SurvExpress database. Please see Figs. 8(A, B).
